# Modulation of the *Erwinia* ligand-gated ion channel (ELIC) and the 5-HT$_3$ receptor via a common vestibule site

Marijke Brams[1], Cedric Govaerts[2], Kumiko Kambara[3], Kerry L Price[4], Radovan Spurny[1], Anant Gharpure[5,6], Els Pardon[7,8], Genevieve L Evans[1], Daniel Bertrand[3], Sarah CR Lummis[4], Ryan E Hibbs[5,6], Jan Steyaert[7,8], Chris Ulens[1]*

[1]Laboratory of Structural Neurobiology, Department of Cellular and Molecular Medicine, Faculty of Medicine, KU Leuven, Leuven, Belgium; [2]Laboratory for the Structure and Function of Biological Membranes, Center for Structural Biology and Bioinformatics, Université libre de Bruxelles, Brussels, Belgium; [3]HiQscreen, Geneva, Switzerland; [4]Department of Biochemistry, University of Cambridge, Cambridge, United Kingdom; [5]Department of Neuroscience, University of Texas Southwestern Medical Center, Dallas, United States; [6]Department of Biophysics, University of Texas Southwestern Medical Center, Dallas, United States; [7]Structural Biology Brussels, Vrije Universiteit Brussel, Brussels, Belgium; [8]VIB-VUB Center for Structural Biology, VIB, Brussels, Belgium

**Abstract** Pentameric ligand-gated ion channels (pLGICs) or Cys-loop receptors are involved in fast synaptic signaling in the nervous system. Allosteric modulators bind to sites that are remote from the neurotransmitter binding site, but modify coupling of ligand binding to channel opening. In this study, we developed nanobodies (single domain antibodies), which are functionally active as allosteric modulators, and solved co-crystal structures of the prokaryote (*Erwinia*) channel ELIC bound either to a positive or a negative allosteric modulator. The allosteric nanobody binding sites partially overlap with those of small molecule modulators, including a vestibule binding site that is not accessible in some pLGICs. Using mutagenesis, we extrapolate the functional importance of the vestibule binding site to the human 5-HT$_3$ receptor, suggesting a common mechanism of modulation in this protein and ELIC. Thus we identify key elements of allosteric binding sites, and extend drug design possibilities in pLGICs with an accessible vestibule site.

*For correspondence:
chris.ulens@kuleuven.be

**Competing interests:** The authors declare that no competing interests exist.

## Introduction

In 1965, Monod, Wyman and Changeux postulated the model of allosteric modulation in proteins (*Monod et al., 1965*). According to this model, proteins exist in two possible conformational states, the tensed (T) or relaxed state (R). The substrate typically has a high affinity for the T state. In multi-subunit proteins, all subunits undergo a concerted transition from the R to the T state upon substrate binding. The equilibrium can be shifted to the R or the T state through a ligand that binds at a site that is different from the substrate binding site, in other words an allosteric site.

Changeux subsequently devoted much of his scientific career to the study of allosteric proteins, with specific attention paid to the nicotinic acetylcholine receptor (nAChR). This protein is a ligand-gated ion channel (LGIC) and thus in effect has no substrate, but the principle of allosteric modulation is similar in that binding of acetylcholine (ACh) shifts the thermodynamic equilibrium from a closed channel state to an open channel state through binding to a site ~50 Å away from the channel

gate. The nAChR is a member of a superfamily of pentameric LGICs (pLGICs), which play major roles in fast synaptic transmission in the central and peripheral nervous systems, and are the site of action of many therapeutic drugs.

Structures of these proteins have been elucidated over the last decade and several nAChR structures are now available for the heteromeric α4β2 and α3β4 nAChRs (*Morales-Perez et al., 2016*; *Walsh et al., 2018*; *Gharpure et al., 2019*), as well as other members of the pLGIC family, including the 5-HT₃ serotonin receptor (*Hassaine et al., 2014*; *Polovinkin et al., 2018*; *Basak et al., 2018*), the glycine receptor (*Du et al., 2015*; *Huang et al., 2015*), the GABA_A receptor (*Zhu et al., 2018*; *Phulera et al., 2018*; *Laverty et al., 2019*; *Masiulis et al., 2019*; *Miller and Aricescu, 2014*), the glutamate-gated chloride channel from *C. elegans* GluCl (*Hibbs and Gouaux, 2011*; *Althoff et al., 2014*) and the prokaryote channels ELIC (*Hilf and Dutzler, 2008*) and GLIC (*Hilf and Dutzler, 2009*; *Bocquet et al., 2009*). Historically, crucial structural insight into the class of nicotinic receptors was derived from cryo-electron microscopic structures of the nAChRs from the electric organ of *Torpedo* (*Miyazawa et al., 2003*; *Unwin, 2005*) and X-ray crystal structures of the acetylcholine binding protein (AChBP; found in certain snails and worms), which is homologous to the extracellular ligand binding domain (LBD) of nAChRs (*Brejc et al., 2001*).

The concept of allosteric modulation is now also more broadly applied to understand the mode of action of certain drugs, called allosteric modulators, which bind at a site that is different from the neurotransmitter binding site, but which can alter energy barriers between multiple conformational states (*Bertrand and Gopalakrishnan, 2007*). For example in the case of pLGICs, positive allosteric modulators (PAMs) of the nAChR can facilitate a transition from a resting to an activated state, thus enhancing the agonist-evoked response. In contrast, negative allosteric modulators (NAMs) hinder such a transition, thus diminishing the agonist response. From a drug development perspective, PAMs or NAMs are highly attractive as they finely tune receptor activation without affecting the normal fluctuations of neurotransmitter release at the synapse. One of the most extensively described PAMs used in the clinic are the benzodiazepines, which act on GABA_A receptors and are widely prescribed as hypnotics, anxiolytics, anti-epileptics or muscle relaxants. Important insights into the molecular recognition of these modulators have been revealed by high-resolution structural data (*Zhu et al., 2018*; *Phulera et al., 2018*; *Laverty et al., 2019*; *Masiulis et al., 2019*; *Miller and Aricescu, 2014*).

With the availability of a growing amount of structural data for these receptors, a diverse array of modulatory molecules have been identified, many of which bind at distinct allosteric binding sites, including general anesthetics (*Nury et al., 2011*; *Sauguet et al., 2013*; *Spurny et al., 2013*), neurosteroids (*Miller et al., 2017*; *Laverty et al., 2017*; *Chen et al., 2018*), lipids (*Zhu et al., 2018*; *Laverty et al., 2019*), antiparasitics (*Hibbs and Gouaux, 2011*; *Althoff et al., 2014*), and many others (*Nys et al., 2016*). Detailed investigation of allosteric sites not only brings further knowledge about the receptor functionality but also uncovers novel drug target sites. However, our current understanding of this multi-site mechanism of allosteric modulation in pLGICs is still incomplete.

In this study, we used complementary structural and functional approaches to expand our understanding of the molecular mechanism of allosteric modulation in pLGICs. Using the prokaryote ELIC channel as a model, we explored the potential of nanobodies (single chain antibodies) as allosteric modulators. We discovered functionally active nanobodies, which act either as a PAM or NAM on ELIC, and determined co-crystal structures to elucidate the nanobody interactions with ELIC. One of the structures reveals an allosteric binding site located near the vestibule of the extracellular ligand-binding domain. Comparison of conservation and divergence in this site in different prokaryotic and eukaryotic receptors suggests a mechanism for achieving subtype-selective allosteric modulation across the receptor superfamily. Using cysteine-scanning mutagenesis and electrophysiological recordings, we show the vestibule site can also be targeted for modulation of the human 5-HT_3A receptor as a proof of principle relevant to other eukaryotic receptors.

## Results and discussion

### Identification of nanobodies active as allosteric modulators on ELIC

In this study, we took advantage of nanobodies, which are high-affinity single chain antibodies derived from llamas; they have been widely employed to facilitate structural studies (*Manglik et al.,*

*2017*) and also hold potential as therapeutics against many possible targets. A first example, caplacizumab (Cabilivi), has recently reached the market (*Scully et al., 2019*). Using the prokaryote ion channel ELIC as a model system, we investigated whether nanobodies could be selected with allosteric modulator activity on ligand-gated ion channels. We expressed ELIC channels in *Xenopus* oocytes and employed automated electrophysiological recordings to characterize a panel of more than 20 different ELIC nanobodies. While none of the nanobodies had any functional effect on ELIC when applied alone, we found that co-application with the agonist GABA evoked a response that broadly falls into three categories. One type of nanobody enhanced the agonist-evoked response, while a second type of nanobody diminished the agonist-evoked response, and the third type had little to no effect. From these classes, we selected several enhancers (PAMs) and inhibitors (NAMs) for a detailed electrophysiological characterization as potential allosteric modulators. In parallel, we conducted X-ray diffraction screening of ELIC plus nanobody co-crystals for structural elucidation. From this selection, we obtained a PAM-active nanobody (PAM-Nb) as well as another NAM-active nanobody (NAM-Nb) and determined their structures bound to ELIC by X-ray crystallography.

Co-application of the agonist GABA with a range of PAM-Nb concentrations demonstrates that PAM-Nb enhances the agonist response (*Figure 1a*) with a $pEC_{50}$-value of $5.37 \pm 0.03$ ($EC_{50}$: 4.2 µM) and $I_{max} = 257 \pm 14\%$ (mean values $\pm$ SEM, n = 8). In contrast, co-application of a range of NAM-Nb concentrations demonstrates that NAM-Nb decreases the agonist response (*Figure 1b*) with a $pIC_{50}$-value of $6.89 \pm 0.03$ ($IC_{50}$: 0.13 µM) and $I_{max} = 34 \pm 2\%$ (n = 6). Unlike competitive antagonists, which fully inhibit the agonist response at saturating concentrations, the inhibition of NAM-Nb levels off at 70% of the response with GABA alone, consistent with the mode of action of certain NAMs. We further investigated the effects of PAM-Nb and NAM-Nb on the GABA concentration-activation curve and obtained results that support their effects as positive and negative allosteric modulators (*Figure 1—figure supplement 1*). These results demonstrate that functionally active nanobodies can be developed against the ligand-gated ion channel ELIC.

## Crystal structures of ELIC in complex with a PAM- or NAM-type nanobody

To gain insight into the structural recognition of positive (PAM-Nb) or negative (NAM-Nb) allosteric modulators, we solved X-ray crystal structures of ELIC in complex with the PAM-Nb or NAM-Nb, respectively (*Supplementary file 1*). The structure of ELIC in complex with the PAM-Nb was determined at a resolution of 2.59 Å, and reveals five PAM-Nb molecules bound to a single pentameric ELIC channel (*Figure 1c,e*). Each PAM-Nb binds at an intrasubunit site in the ELIC extracellular ligand binding domain. When viewed from the top along the fivefold symmetry axis, the nanobodies extend outward and the structure resembles a five-bladed propeller (*Figure 1c*). The structure of ELIC in complex with NAM-Nb was determined at a resolution of 3.25 Å and is structurally distinct from the complex with PAM-Nb: instead of five Nb molecules bound to the ELIC pentamer, here a single NAM-Nb molecule is bound to the ELIC pentamer (*Figure 1d,f*). The NAM-Nb binds at the channel vestibule entrance and near to the N-terminal α-helix of a single ELIC subunit, and is oriented in such a manner that only a single nanobody molecule can bind at this interface, as the core of the nanobody sterically hinders access to the four other sites (*Figure 1d,f*).

A more detailed analysis of the interaction interface between both nanobodies and ELIC reveals remarkable features (*Figure 2*). The PAM-Nb binds to the extracellular ligand-binding domain and forms extensive interactions through the complementarity determining region CDR1 (residues S25-I33) of the nanobody (*Figure 2a*). The tip of the finger of this loop region wedges in between the ELIC β8- and β9-strand, forming a distinct anti-parallel β-sheet interaction with the β8-strand. The CDR1 loop region points toward an allosteric binding site previously identified in a chimera of the human α7 nAChR and the acetylcholine binding protein, α7-AChBP (*Li et al., 2011*) (see complex with fragment molecule CU2017, pdb code 5oui) (*Figure 2a*). In other words, the PAM-Nb binds to a site in ELIC that corresponds to a functionally important allosteric site in the human α7 nAChR, consistent with its function as an allosteric modulator.

The mode of interaction of the NAM-Nb with ELIC is distinct from the PAM-Nb (*Figure 2b*). The interaction interface is remarkable in that a unique α-helical region within the CDR3-region (P104-L110) forms the interface with ELIC. This α-helical region is positioned perpendicularly upon the α′1-helix in ELIC (N60-N69) and forms the outer rim of an allosteric binding site previously identified as the vestibule binding site (*Spurny et al., 2012*). This site is also the target for the benzodiazepine

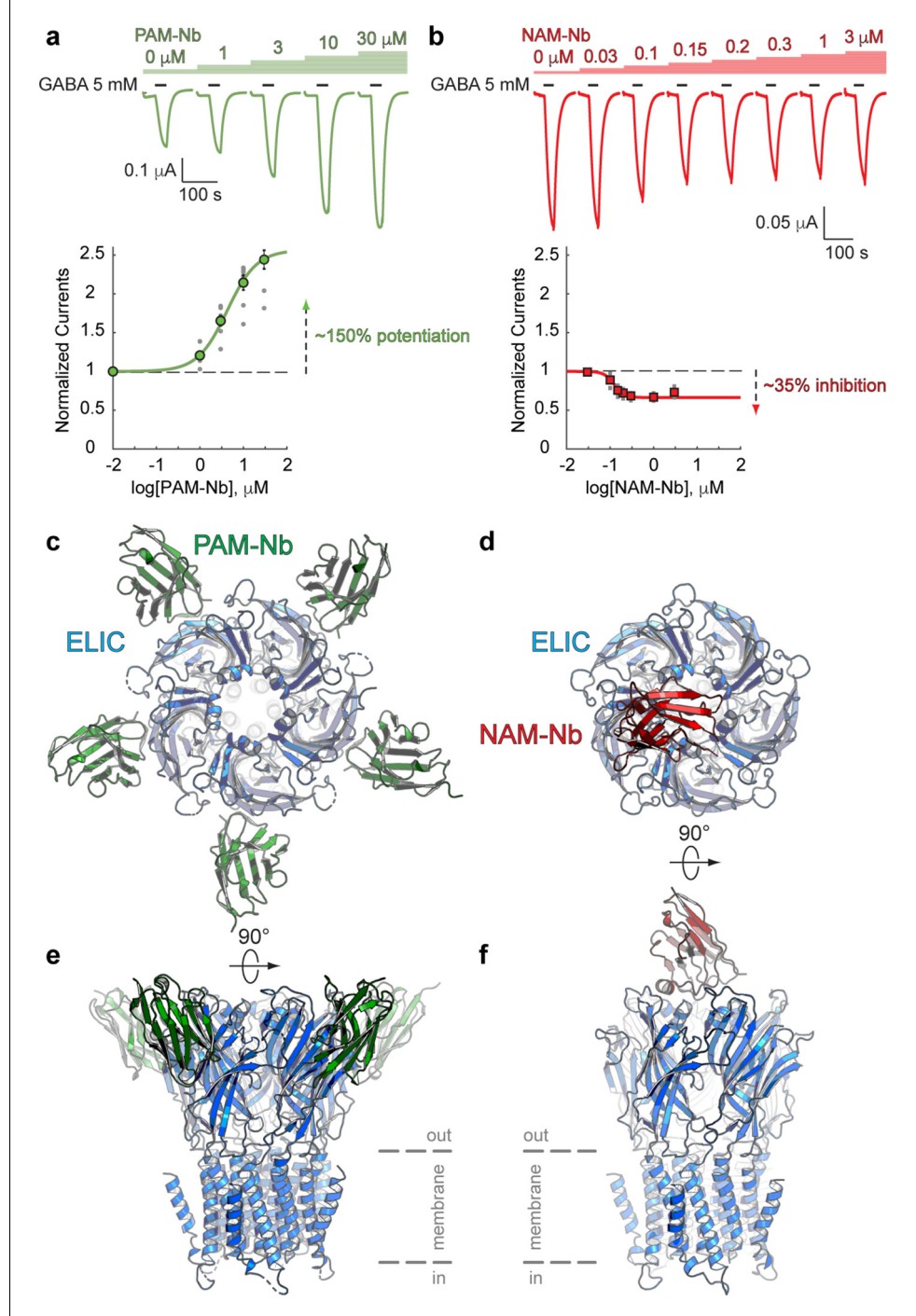

**Figure 1.** Nanobodies active as allosteric modulators and structures bound to the ELIC channel. (a, b) Electrophysiological recordings of ELIC activated by the agonist GABA and in the presence of increasing concentrations of PAM-Nb (a, green) or NAM-Nb (b, red). The curve represents a fit to the Hill equation to the normalized current responses. Circles represent averaged data with standard errors. (c,d) X-ray crystal structures of ELIC bound by PAM-Nb (c) or NAM-Nb (d). The cartoon representation shows a top-down view onto the ELIC pentamer along the fivefold axis (blue). (e,f) Side views from c,d. The dashed lines indicate the presumed location of the membrane boundaries.

The online version of this article includes the following source data and figure supplement(s) for figure 1:

**Source data 1.** Electrophysiological recordings of ELIC with PAM-Nb and NAM-Nb.

**Figure supplement 1.** The effect of PAM-Nb and NAM-Nb on the GABA concentration-activation curve.

*Figure 1 continued on next page*

*Figure 1 continued*

**Figure supplement 1—source data 1.** ELIC concentration-activation curves in the presence of PAM-Nb and NAM-Nb.
**Figure supplement 2.** Simulated annealing omit map for PAM-Nb ELIC structure.
**Figure supplement 3.** Simulated annealing omit map for NAM-Nb ELIC structure.

flurazepam, which acts as a positive allosteric modulator on ELIC (*Spurny et al., 2012*) (*Figure 2b* inset). This site is distinct from the high-affinity benzodiazepine site at the α/γ subunit interface (*Masiulis et al., 2019*). Similar to the PAM-Nb, it is interesting to observe that the allosteric binding

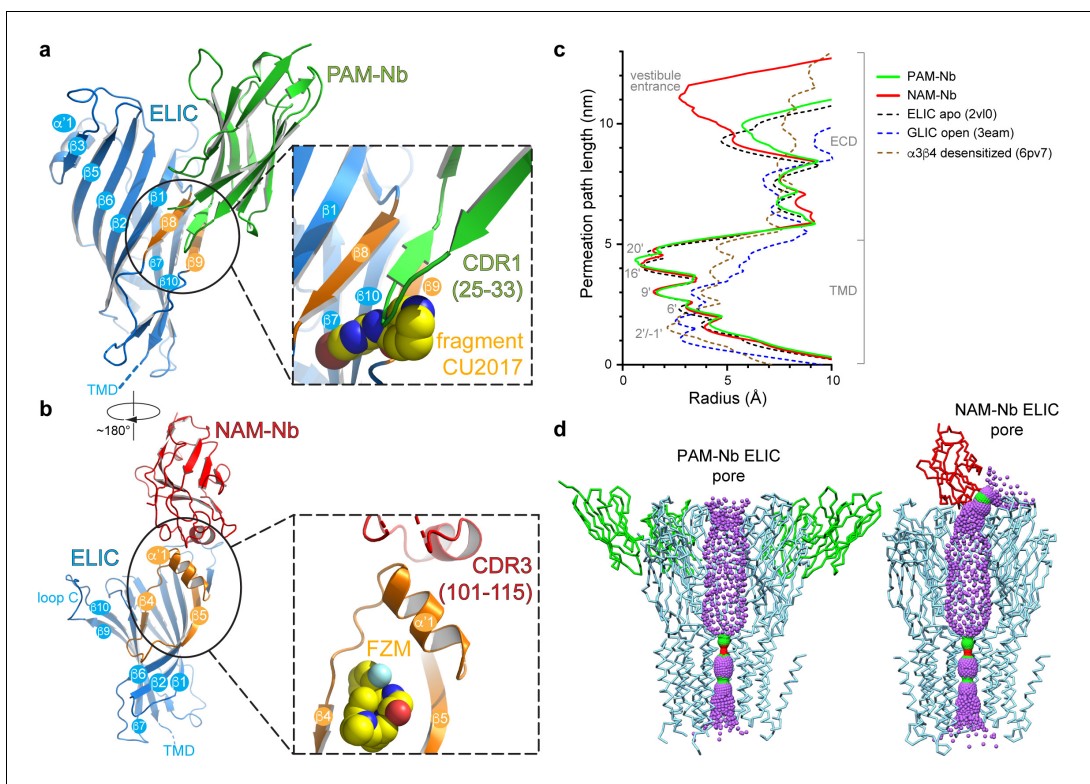

**Figure 2.** Detailed nanobody interaction sites in ELIC and channel pore analysis. (**a**) A detailed view of the interaction between PAM-Nb (green) and a single ELIC subunit (blue). The binding site for PAM-Nb overlaps with a known allosteric binding site, in a related pLGIC, for a small molecule fragment called CU2017, shown as spheres (carbon yellow, nitrogen blue) (*Delbart et al., 2018*; pdb code 5oui). (**b**) Detailed view of the interaction between NAM-Nb (red) and a single ELIC subunit (blue). The binding site for NAM-Nb involves a region which borders a known allosteric binding site for flurazepam in ELIC (*Spurny et al., 2012*; pdb code 2yoe). (**c,d**) Analysis of the ELIC channel pore radius for PAM-Nb, NAM-Nb bound structures and references structures.
The online version of this article includes the following source data and figure supplement(s) for figure 2:

**Source data 1.** Analysis of pore radius profiles.
**Figure supplement 1.** Detailed atomic interactions between PAM-Nb and ELIC, compared to those between CU2017 (a NAM molecule) and α7-AChBP (*Delbart et al., 2018*).
**Figure supplement 1—source data 1.** List of amino acid interactions in the PAM-Nb bound ELIC structure versus CU2017 bound alpha7-AChBP structure.
**Figure supplement 2.** Detailed atomic interactions between NAM-Nb and ELIC, compared to those between flurazepam (a PAM molecule) and ELIC (*Spurny et al., 2012*).
**Figure supplement 2—source data 1.** List of amino acid interactions in the NAM-Nb bound ELIC structure versus flurazepam bound ELIC structure.
**Figure supplement 3.** Analysis of pore radius profiles through lateral fenestrations located at subunit interfaces in ELIC and the β3 GABA$_A$R (*Miller and Aricescu, 2014*).

site of the NAM-Nb is adjacent to a previously identified binding site for a small molecule allosteric modulator. Remarkably, in each case, the nanobody has the opposite functional effect of the small molecule at the same site. The PAM-Nb acts as a positive modulator, whereas the CU2017 fragment acts as a negative modulator at the β8-β9 site. Conversely, the NAM-Nb acts as a negative modulator, whereas flurazepam acts as a positive modulator at the vestibule site. This result demonstrates that the same allosteric site can be targeted both by a PAM or NAM, and that its functional action likely depends on defined side chain interactions. This is in agreement with previous pharmacological studies, which have shown that a substitution as small as a methylation of an aromatic ring in a small molecule modulator can alter the functional profile from a PAM to a NAM of the α7 nAChR (*Gill-Thind et al., 2015*). To further explore positive versus negative modulation, the atomic interactions of the PAM-Nb or NAM-Nb with ELIC are compared with those formed by the small molecules CU2017 or flurazepam, respectively (see *Figure 2—figure supplements 1–2*). The results from this analysis reveal some residues that are uniquely targeted in the PAM-Nb versus the CU2017 complex, which could be responsible for positive versus negative allosteric modulation (*Figure 2—figure supplement 1*). Similarly in the NAM-Nb versus the flurazepam complex, adjacent subsets of residues in the vestibule site are targeted by the different molecules, again indicating that distinct residues within the same region play a role in negative versus positive allosteric modulation (*Figure 2—figure supplement 2*).

The pores of both Nb complexes resemble previous structures of ELIC, with narrow constriction points at the 9', 16', and 20' positions (*Hilf and Dutzler, 2008*), suggesting that the channels are in non-conducting conformations (*Figure 2c,d*). This closed state is structurally distinct from the putative desensitized state of the nAChR, whose primary gate lies at the cytoplasmic end at the −1' position (*Morales-Perez et al., 2016*; *Gharpure et al., 2019*). Further analysis of the central channel axis of the NAM-Nb complex reveals a partial blockade of the extracellular vestibule entrance by the nanobody, resulting in a radius profile comparable to the 6' pore constriction (*Figure 2c,d*). This partial constriction of the ion permeation pathway could possibly explain why the NAM-Nb only partially inhibits GABA-induced currents. Alternatively, it is possible that in ELIC alternate pathways exist for ion entry through lateral fenestrations located at subunit interfaces, as seen in other members of the pLGIC family (*Zhu et al., 2018*; *Miller and Aricescu, 2014*) (*Figure 2—figure supplement 3*). Finally, NAM-Nb binding could restrict flexibility in the extracellular domain (ECD), thereby limiting a transition from a closed to a conductive conformation (*Sauguet et al., 2014*).

## Subtype-dependent vestibule site access in different prokaryote and eukaryote receptors

To further investigate the possible conservation of the vestibule binding site in different prokaryote and eukaryote pLGICs, we performed a systematic analysis of the vestibule site architecture in all currently available pLGIC structures. The results from this analysis show that the outer rim of the vestibule site, which corresponds to residues N60-F95 in ELIC, can adopt one of three possible conformations (*Figure 3*).

In certain structures, we observe that the stretch of amino acids between the β4- and β5-strands resembles the shape of the Greek letter Ω, for example in the α4 nAChR subunit (*Morales-Perez et al., 2016*; *Walsh et al., 2018*), and therefore this region is termed the Ω loop (*Hu et al., 2018*). In *Figure 3a*, the outer rim of the vestibule site in the α4 nAChR is shown in blue (from the helix to the β5-strand) and the Ω loop in red. The tip of the Ω loop can either point into the vestibule as in the α4 nAChR subunit (*Morales-Perez et al., 2016*; *Walsh et al., 2018*), which we call the Ω-in conformation (*Figure 3a*), or the tip of the Ω loop can point outward, which we call the Ω-out conformation, for example in the α1 GlyR subunit (*Du et al., 2015*) (outer rim shown in blue, Ω-out in green, *Figure 3b*). Importantly, in both of these conformations access to the vestibule site is occluded. In the Ω-in conformation, the tip of the Ω loop prevents access to the vestibule site of its own α4 nAChR-subunit (*Figure 3a,d*), whereas in the Ω-out conformation vestibule access is prevented by the tip of the Ω-out loop of its neighboring α1 GlyR (-) subunit (*Figure 3b,e*). In addition to the Ω-in and Ω-out conformations, we observe a third possible conformation in which the Ω loop is stretched, for example in the 5-HT$_{3A}$R (*Hassaine et al., 2014*) (*Figure 3c*) and ELIC, and creates an accessible vestibule site (*Figure 3f*). Consequently, we term this conformation the Ω-open conformation (Ω-open is shown in magenta, *Figure 3c,f*).

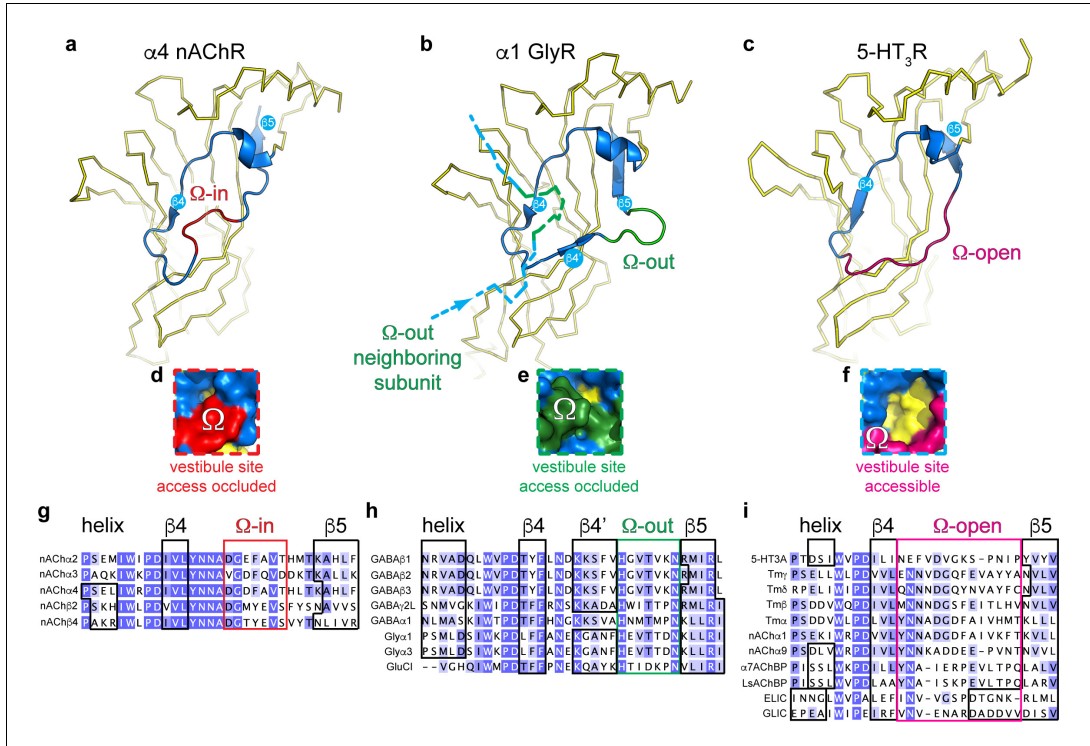

**Figure 3.** Distinct conformations of the vestibule site in pentameric ligand-gated ion channels. (**a–c**) Yellow ribbon representation of a single subunit ligand binding domain. Part of the vestibule site (shown in blue cartoon), called the Ω-loop, adopts three distinct conformations in different pLGICs: the Ω-in (red, (**a**), Ω-out (green, (**b**) and Ω-open conformation (magenta, (**c**). (**d-f**) Insets show a zoom of the Ω-loop in surface representation to illustrate occluded vestibule site access in the Ω-in and Ω-out conformations, compared to an accessible vestibule site in the Ω-open conformation. (**g–i**). Sequence alignment of the Ω-loop and neighboring residues in pLGICs for which structures have been elucidated.

The online version of this article includes the following figure supplement(s) for figure 3:

**Figure supplement 1.** B-factor analysis of the Ω-loop compared to other regions within each structure.

A systematic comparison of the Ω loop conformations demonstrates that the different pLGIC sub-families group into defined categories (reference structures are listed in *Supplementary file 2*). First, we observe that all currently known anionic receptor structures adopt the Ω-out conformation (*Figure 3h*). This implies that the vestibule access is occluded in all of these receptors, for example in the homomeric α1 GlyR (*Du et al., 2015*), α3 GlyR (*Huang et al., 2015*), or GluCl (*Hibbs and Gouaux, 2011*) as well as the heteropentameric α1βγ2 GABA_ARs (*Zhu et al., 2018*; *Phulera et al., 2018*; *Laverty et al., 2019*) and homopentameric β3 GABA_AR (*Miller and Aricescu, 2014*). The Ω-out loop sequence is strongly conserved in these receptors, with a His residue and Asn residue at either end of the Ω loop and a Thr at the tip (not conserved in GluCl). We also observe that in all these cases the Ω loop is preceded by a stretch of amino acids that forms an additional β-strand, which we call the β4'-strand, as it also follows the β4-strand. The β4'-strand contains a well conserved start Lys residue, which is present in GlyRs, GABA_ARs and GluCl. The sequence conservation in these two regions suggests an important functional role. In contrast with anionic receptors, we observe that certain cationic receptors can adopt either the Ω-in or Ω-open conformation. However, none of the cationic receptors adopt the Ω-out conformation. Only 5 nAChR subunits adopt the Ω-in conformation, namely the α2 nAChR (*Kouvatsos et al., 2016*), α4 and β2 nAChR-subunits (*Morales-Perez et al., 2016*; *Walsh et al., 2018*) and α3 and β4 nAChR-subunits (*Gharpure et al., 2019*). This implies that the heteropentameric α4β2 and α3β4 nAChRs adopt an all-subunit-occluded vestibule site conformation. The stretch of amino acids that form the Ω-in conformation is also well conserved with a Gly-Val on either end and an aromatic residue (Phe/Tyr) at the tip, again suggesting an important functional role. In contrast, all other nAChR subunits (*Dellisanti et al., 2007*; *Zour et al., 2014*),

including the *Torpedo* nAChR (*Miyazawa et al., 2003*), as well as the 5-HT$_{3A}$R (*Hassaine et al., 2014*), snail AChBPs (*Brejc et al., 2001*; *Celie et al., 2005*), prokaryote GLIC (*Bocquet et al., 2009*; *Hilf and Dutzler, 2009*) and ELIC adopt the Ω-open conformation. The sequence of amino acids forming the Ω-open loop is less conserved, except for the position +two following the β4-sheet, which is a Asn in all receptors except 5-HT$_{3A}$R. Important to note is that the Ω-open loop in the α9 nAChR (*Zour et al., 2014*) is disordered in the apo state (pdb code 4d01), but not in the antagonist (MLA)-bound state (pdb code 4uxu). This raises the intriguing possibility that the Ω loop is conformationally flexible and could control access to the vestibule site. A comprehensive analysis of all pLGIC structures available to date further revealed that in certain structures the average B-factors per residue, which are used as an indicator of vibrational movement, are enhanced in the Ω-loop relative to other parts of the structure (*Figure 3—figure supplement 1*). Some of the most striking examples are the Ω-in loop of the α3 subunit in the α3β4 nAChR (pdb code 6pv7, desensitized state) (*Gharpure et al., 2019*), the Ω-out loop in POPC-bound GluCl (pdb code 4tnw, lipid-modulated state) (*Althoff et al., 2014*) and the Ω-open loop in GLIC T25′A (pdb code 4lmj, liganded closed state) (*Gonzalez-Gutierrez et al., 2013*). These examples also represent possible intermediate or end states of the gating cycle, suggesting that the Ω-loop could show enhanced movement during the gating reaction. A functional role of the Ω-loop in gating of other pLGIC was demonstrated in earlier studies using a photochemical proteolysis approach of the Ω-loop in GABA$_A$Rs (*Hanek et al., 2010*), incorporation of flexible glycine linkers in GABA$_A$Rs (*Venkatachalan and Czajkowski, 2012*) and rate-equilibrium free energy relationship analysis of Ω-loop mutants of muscle nAChRs (*Chakrapani et al., 2003*).

Together, the results from this analysis demonstrate that different receptor subtypes adopt a different Ω loop conformation, with the Ω-open conformation creating an accessible vestibule site, whereas the Ω-in and Ω-out conformations occlude vestibule site access. Additionally, in heteropentameric GABA$_A$ receptors the vestibule is occupied by N-linked glycan chains (*Zhu et al., 2018*; *Phulera et al., 2018*; *Laverty et al., 2019*), thus creating an additional level of restriction on vestibule site access in these and possibly other pLGICs. Our analyses provide new opportunities for drug design of allosteric molecules that have subtype-specific pharmacology (due to low sequence conservation compared to the orthosteric site), based on vestibule site access.

## Cysteine-scanning mutagenesis in the vestibule site of the human 5-HT$_{3A}$ receptor

Based on our observations that the vestibule site in ELIC is the target for positive allosteric modulators such as the benzodiazepine flurazepam (*Spurny et al., 2012*) as well as negative allosteric modulators such as the NAM-nanobody described in this study, we further investigated whether the mechanism of vestibule site modulation is functionally conserved in eukaryote receptors. We chose the 5-HT$_{3A}$R as a prototype receptor since it has a clear Ω-open vestibule conformation and its structure (*Hassaine et al., 2014*) as well as functional and pharmacological properties (*Lummis, 2012*) are well described. We then employed the substituted cysteine accessibility method (SCAM) (*Karlin and Akabas, 1998*) to investigate the functional effects on channel gating before and after modification of cysteines in the 5-HT$_{3A}$R vestibule site with the cysteine-reactive reagent MTSEA-biotin, which is approximately the size of a small molecule modulator such as flurazepam. We chose residues on the outer rim of the vestibule site, T112 (top) and F125 (bottom), respectively, as well as residues deeper into the vestibule site, N147, K149 and L151 on the β6-strand and Y86 on the β2-strand (*Figure 4b*). It is interesting to note that N147, K149 and L151 are located on the opposite side of the β-strand to the loop E residues (Q146 and Y148), while Y86 is on the opposite side of the β-strand to the loop D residues (W85 and R87); both of these regions are functionally important contributors to the neurotransmitter binding site (*Hassaine et al., 2014*). Residue P111, which points away from the vestibule site, was included as a negative control.

Representative current traces of channel responses to increasing concentrations of serotonin (5-HT) are shown in *Figure 4a* for the L151C mutant before and after modification with MTSEA-biotin. The channel responses are drastically increased after cysteine modification, which is caused by a more than 10-fold increase of the maximal current response (I$_{max}$) of the concentration-activation curve after MTSEA-biotin modification as well as a shift of the EC$_{50}$-value to lower concentrations (*Figure 4a,c*) (pEC$_{50}$: 4.01 ± 0.03, n = 5 (EC$_{50}$: 98 μM) versus pEC$_{50}$: 4.96 ± 0.04, n = 5 (EC$_{50}$: 11 μM)), indicating a strong PAM-effect at this position (p<0.0001, mean values ± SEM). The L151C

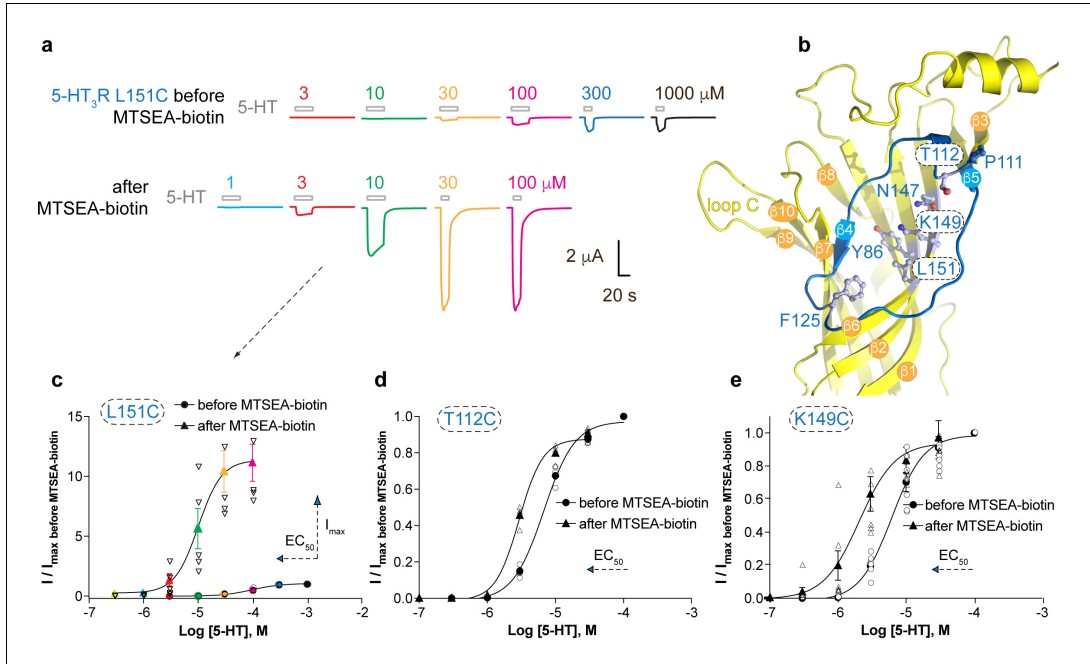

**Figure 4.** Allosteric modulation of the 5-HT$_3$A receptor through chemical modification of engineered cysteines in the vestibule site. (a) Example traces of agonist-evoked channel responses of the L151C 5-HT$_{3A}$R mutant before and after modification with MTSEA-biotin show potentiation after cysteine modification. (b) Location of L151C and other engineered cysteine mutants in this study, shown in ball and stick representation. (c) Concentration-activation curves before and after modification with MTSEA-biotin are a Hill curve fit for the recordings shown in (a) as well as additional data for T112C (d) and K149C (e). (c–e) Each of these three mutants reveal a leftward shift of the curve upon cysteine modification, consistent with a positive allosteric effect. In the case of L151C, this effect is combined with a large increase of the maximal current response.

The online version of this article includes the following source data for figure 4:

**Source data 1.** Electrophysiological recordings of 5-HT$_3$R mutants before and after treatment with MTSEA-biotin.

mutation alone causes a ~50 fold increase of the EC$_{50}$-value compared to wild type (EC$_{50}$ 98 μM versus 1.8 μM), suggesting that this mutation impacts function, possibly because L151 contributes to a hydrophobic patch in the center of the subunit (it is <4 Å from I100 and L131), and this, or the mutation itself, could affect the adjacent Cys-loop and loop B. Modification of this residue with MTSEA-biotin appears to change the EC$_{50}$ (11 μM) back toward wild-type value (1.8 μM). Mutants T112C and K149C also showed a lesser but significant decrease of the EC$_{50}$-value after MTSEA-biotin, namely 6.7 μM (pEC$_{50}$: 5.17 ± 0.02, n = 4) versus 2.9 μM (pEC$_{50}$: 5.53 ± 0.02, n = 4) for T112C (*Figure 3d*) (p<0.0001) and 6.2 μM (pEC$_{50}$: 5.21 ± 0.05, n = 7) versus 2.1 μM (pEC$_{50}$: 5.69 ± 0.11, n = 7) for K149C (*Figure 3e*) (p=0.005). Although neither of these mutants displayed an increase of I$_{max}$ as in the L151C mutant, the leftward shift of the concentration-activation curve is also consistent with a PAM-effect at these positions. No significant differences were observed for current responses before and after MTSEA-biotin for Y86C, F125C and N147C or the negative control, P111C (see *Supplementary file 3*), suggesting modification of these residues does not place the MTSEA-biotin in an appropriate position to act as a modulator, although it is also possible it did not react here. Together, these results demonstrate that general perturbation of this area via a drug binding or a mutation can alter receptor activation and that allosteric modulation via the vestibule binding site is functionally conserved between ELIC and 5-HT$_{3A}$ receptors. The evolutionary distance between these receptors, combined with previous work, suggests that this mechanism of modulation possibly also extends to other pLGIC members. In support of this hypothesis the functional importance of the vestibule site in channel modulation is now emerging from a wide range of different structural studies on prokaryote channels, demonstrating ELIC modulation by flurazepam (*Spurny et al., 2012*) or nanobodies (this study), 4-bromocinnamate modulation of the prokaryote homolog sTeLIC (*Hu et al., 2018*), and acetate binding in GLIC (*Fourati et al., 2015*). Additionally, the vestibule site

has been identified as the site of modulation in different eukaryote channels using various approaches, including 5-HT$_{3A}$R modulation by cysteine-scanning mutagenesis (this study), α7 nAChR modulation using a fragment-based screening approach (*Spurny et al., 2015*; *Delbart et al., 2018*), and Zn$^{2+}$-mediated inhibition of the α1 GlyR (*Miller et al., 2008*).

## Conclusion

In conclusion, we demonstrate here that functionally active nanobodies can be developed for the ligand-gated ion channel ELIC. Functional characterization demonstrates that nanobodies can act as positive or negative allosteric modulators on this channel. Previous structural studies on the mouse 5-HT$_3$ receptor have also identified a nanobody with an inhibitory mode of action (*Hassaine et al., 2014*), while a nanobody active as a positive allosteric modulator was recently developed for the human α1β3γ2 GABA$_A$ receptor (*Miller et al., 2018*). These results indicate that functionally active nanobodies can be developed for different prokaryote and eukaryote ligand-gated ion channels and that they are useful tools for structural studies.

Crystal structures reveal that ELIC nanobodies can interact via distinct epitopes, including accessible parts of allosteric binding sites previously discovered in the extracellular ligand-binding domain and bound by small molecules. One potentially attractive site for further development of allosteric modulators is the vestibule site, which can be targeted not only with nanobodies or small molecules in ELIC, but also by chemical modification of engineered cysteines as demonstrated in the human 5-HT$_{3A}$ receptor. The vestibule site offers opportunities to further develop both positive and negative modulators, as well as to exploit subtype-specific access to certain receptors. These results pave the way for the future development of novel therapeutics that can modulate channel activity in pLGIC-related disorders. An attractive path would be to expand the currently available repertoire of therapeutics with pharmacologically active nanobodies against human pLGICs.

# Materials and methods

**Key resources table**

| Reagent type (species) or resource | Designation | Source or reference | Identifiers | Additional information |
|---|---|---|---|---|
| Gene | *Erwinia* ligand-gated ion channel (ELIC) | Synthetic gene from Genscript | UniProt P0C7B7 | |
| Gene | Human 5-hydroxytryptamine receptor 3A (5-HT$_{3A}$ R) | John Peters laboratory (*Belelli et al., 1995*) | GenBank NM_000869 | |
| Cell line | C43 *E. coli* strain | Lucigen | #60446 | |
| Cell line | WK6 *E. coli* strain | *Zell and Fritz, 1987* | | |
| Cell line | TG1 *E. coli* strain | Lucigen | #60502 | |
| Antibody | PAM-Nb | | | Nanobody obtained by immunizing an adult *llama glama* with recombinant ELIC protein. PAM-Nb was used at 1–30 µM concentration in electrophysiology experiments. In protein crystals the PAM-Nb concentration is > 1 mM. |
| Antibody | NAM-Nb | | | Nanobody obtained by immunizing an adult *llama glama* with recombinant ELIC protein. NAM-Nb was used at 0.03–3 µM concentration in electrophysiology experiments. In protein crystals the NAM-Nb concentration is > 1 mM. |
| Recombinant DNA reagent | pGEM-HE | *Liman et al. (1992)* | | |

*Continued on next page*

*Continued*

| Reagent type (species) or resource | Designation | Source or reference | Identifiers | Additional information |
|---|---|---|---|---|
| Recombinant DNA reagent | pMESy4 | | GenBank KF415192 | |
| Sequence-based reagent | P111C oligonucleotide forward and reverse | | | 5'-caccaagttgtccatcTG cacggacagcatctgg-3' 5'-ccagatgctgtccgtgCA gatggacaacttggtg- 3' |
| Sequence-based reagent | T112C oligonucleotide forward and reverse | | | 5'-caagttgtccatccccTG Cgacagcatctgggtcc-3' 5'-ggacccagatgctgtcGC Aggggatggacaacttg-3' |
| Sequence-based reagent | N147C oligonucleotide forward and reverse | | | 5'-caaggcgaagttcagTG ctacaagccccttcagg-3' 5'-cctgaaggggcttgtagC Actgaacttcgccttg-3' |
| Sequence-based reagent | K149C oligonucleotide forward and reverse | | | 5'-ggcgaagttcagaactac TGCccccttcaggtggtga-3' 5'- tcaccacctgaaggggGC Agtagttctgaacttcgcc-3' |
| Sequence-based reagent | L151C oligonucleotide forward and reverse | | | 5'- gttcagaactacaagccc TGtcaggtggtgactgc-3' 5'-gcagtcaccacctgaCAg ggcttgtagttctgaac-3' |
| Sequence-based reagent | Y86C oligonucleotide forward and reverse | | | 5'- ctggtaccggcagtGct ggactgatgag-3' 5'-ctcatcagtccagCactg ccggtaccag-3' |
| Sequence-based reagent | F125C oligonucleotide forward and reverse | | | 5'-ggacattctcatcaatgagt Gcgtggatgtggg-3' 5'-cccacatccacgCactcatt gatgagaatgtcc-3' |
| Sequence-based reagent | Primer for nanobody library generation CALL001 | *Pardon et al. (2014)* | | 5'-GTCCTGGCTGCTC TTCTACAAGG-3' |
| Sequence-based reagent | Primer for nanobody library generation CALL002 | *Pardon et al. (2014)* | | 5'-GGT ACGTGCTGT TGAACTGTTCC-3' |
| Sequence-based reagent | Primer for nanobody library generation VHH-Back | *Pardon et al. (2014)* | | 5'-GATGTGCAGCTGCAG GAGTCTGGRGGAGG-3'(PstI) |
| Sequence-based reagent | Primer for nanobody library generation VHH-For | *Pardon et al. (2014)* | | 5'-CTAGTGCGGCCGCTGG AGACGGTGACCTGGGT-3'(Eco91I) |
| Sequence-based reagent | Primer for nanobody library analysis MP57 | *Pardon et al. (2014)* | | 5'-TTATGCTTCCGGCTC GTATG-3' |
| Peptide, recombinant protein | Primer for nanobody library analysis GIII | *Pardon et al. (2014)* | | 5'-CCACAGACAGCCCTCATAG-3' |
| Commercial assay or kit | mMessage mMachine T7 Transcription kit | ThermoFisher Scientific | #AM1344 | |
| Commercial assay or kit | Ni Sepharose 6 Fast Flow resin | GE Healthcare | #17531802 | |
| Commercial assay or kit | Superdex 75 10/300 GL column | GE Healthcare | # 17-5174-01 | |
| Commercial assay or kit | Superdex 200 Increase 10/300 GL | GE Healthcare | # 28990944 | |

*Continued on next page*

*Continued*

| Reagent type (species) or resource | Designation | Source or reference | Identifiers | Additional information |
|---|---|---|---|---|
| Commercial assay or kit | QuikChange Site-Directed Mutagenesis Kit | Agilent | # 200518 | |
| Chemical compound, drug | 5-HT creatinine sulphate | Sigma-Aldrich | #S2805 | |
| Chemical compound, drug | MTSEA-biotin | Biotium | #90064 | |
| Chemical compound, drug | GABA | Sigma-Aldrich | #A2129 | |
| Chemical compound, drug | *E. coli* total lipid extract | Avanti Polar Lipids | #100500P | |
| Chemical compound, drug | Anagrade n-undecyl-β-D-maltoside (UDM) | Anatrace | #U300 | |
| Software, algorithm | GraphPad Prism Software v4.03 | | RRID:SCR_002798 | |
| Software, algorithm | CCP4 | *Winn et al., 2011* | | |
| Software, algorithm | STARANISO | *Tickle et al., 2018* http://staraniso.globalphasing.org | | |
| Software, algorithm | XDS | *Kabsch, 2010* | | |
| Software, algorithm | Coot | *Emsley et al. (2010)* | | |
| Software, algorithm | Buster | *Smart et al. (2012)* | | |
| Software, algorithm | PDB-REDO | *Joosten et al. (2014)* | | |
| Software, algorithm | MolProbity | *Chen et al. (2010)* | | |
| Software, algorithm | PyMOL v2.3.0 | Schrödinger | RRID:SCR_000305 | |
| Software, algorithm | HOLE | *Smart et al. (1996)* | | |
| Software, algorithm | CAVER | *Jurcik et al. (2018)* | | |
| Software, algorithm | PHENIX | *Adams et al. (2010)* | | |
| Software, algorithm | HiClamp | Multi Channel Systems | | |

## Production of nanobodies against ELIC

A llama was immunized with 2 mg in total of purified wild type ELIC protein over a period of 6 weeks using a previously established protocol (*Pardon et al., 2014*). Briefly, from the anti-coagulated blood of the immunized llama, lymphocytes were used to prepare cDNA. This cDNA served as a template to amplify the open reading frames coding for the variable domains of the heavy-chain only antibodies, also called nanobodies. The PCR fragments were ligated into the pMESy4 phage display vector and transformed in *E. coli* TG1 cells. A nanobody library of $1.1 \times 10^8$ transformants was obtained. For the discovery of ELIC-specific nanobodies, wild-type ELIC was solid phase coated directly on plates and phages were recovered by limited trypsinization. After two rounds of selection, periplasmic extracts were made and subjected to ELISA screens, seven different families were confirmed by sequence analysis. All clones were produced and purified as previously described (*Pardon et al., 2014*).

## Nanobody expression and purification

A series of nanobodies were individually expressed in the periplasm of *E. coli* strain WK6, which was grown in TB media supplemented with 0.1 mg/ml carbenicillin, 0.1% glucose and 2 mM MgCl$_2$ to an absorbance A$_{600}$ ~0.7 at 37°C. The culture was induced with 1 mM isopropyl β-D-1-thiogalactopyranoside (IPTG) and incubated in an orbital shaker overnight at 28°C. Cells were harvested by centrifugation, resuspended in TES buffer (200 mM TRIS, pH 8.0; 0.5 mM EDTA; 500 mM sucrose supplemented with 40 mM imidazole) and incubated for 1 hr. To this fraction four times diluted TES buffer was added and incubated for 1 hr. This fraction was cleared by centrifugation at 10,000 g. The supernatant was incubated with Ni Sepharose 6 Fast Flow resin (GE Healthcare) and incubated for 1 hr at room temperature. The beads were washed with buffer containing 20 mM TRIS pH 8.0, 300 mM NaCl and 40 mM imidazole. Protein was eluted with the same buffer supplemented with 300 mM imidazole. The eluted protein was concentrated to less than 1 ml on a 3 kDa cut-off Vivaspin concentrating column (Sartorius) and further purified on a Superdex 75 10/300 GL column (GE Healthcare) equilibrated with 10 mM Na-phosphate (pH 8.0) and 150 mM NaCl. Peak fractions corresponding to nanobody were pooled and spin-concentrated to ~50 mg/ml.

## Automated voltage-clamp recordings of ELIC

For expression of ELIC in *Xenopus* oocytes, we used the pGEM-HE expression plasmid (*Liman et al., 1992*) containing the signal sequence of the human α7 nAChR followed by the mature ELIC sequence, as previously described (*Spurny et al., 2012*). After plasmid linearization with *NheI*, capped mRNA was transcribed in vitro using the mMESSAGE mMACHINE T7 transcription kit (ThermoFisher). 2 ng of mRNA per oocyte was injected into the cytosol of stage V and VI oocytes using the Roboinject automated injection system (Multi Channel Systems). Oocyte preparations and injections were done using standard procedures (*Knoflach et al., 2018*). Injected oocytes were incubated in ND96-solution containing 96 mM NaCl, 2 mM KCl, 1.8 mM CaCl$_2$, 2 mM MgCl$_2$ and 5 mM HEPES, pH 7.4, supplemented with 50 mg/L gentamicin sulfate. One to five days after injection, electrophysiological recordings were performed at room temperature by automated two-electrode voltage clamp with the HiClamp apparatus (Multi Channel Systems). Cells were superfused with standard OR2 solution containing 82.5 mM NaCl, 2.5 mM KCl, 1.8 mM CaCl$_2$, 1 mM MgCl$_2$, 5 mg/L BSA and 5 mM HEPES buffered at pH 7.4. Cells were held at a fixed potential of −80 mV throughout the experiment. Agonist-evoked current responses were obtained by perfusing oocytes with a range of GABA concentrations in OR2 solution. Different nanobodies diluted into OR2 solution were tested at a range of concentrations by pre-incubation with nanobody alone and followed by a co-application of nanobody with 5 mM GABA. Data acquired with the HiClamp were analyzed using the manufacturer's software (Multi Channel Systems). Concentration-activation curves were fitted with the empirical Hill equation as previously described (*Spurny et al., 2012*). Data are presented as the mean ± standard error of the mean (SEM) with the raw data points overlaid as a dot plot in the relevant figures.

## ELIC purification and crystallization of ELIC-nanobody complexes

Purified ELIC protein was produced as previously described, but with minor modifications (*Spurny et al., 2012*). In brief, the ELIC expression plasmid was transformed into the C43 *E. coli* strain and cells were grown in LB medium. Protein expression was induced with 200 µM isopropyl β-D-1-thiogalactopyranoside (IPTG) and incubated in an orbital shaker at 20°C overnight. After cell lysis, membranes were collected by ultracentrifugation at 125,000 x g and solubilized with 2% (w/v) anagrade n-undecyl-β-D-maltoside (UDM, Anatrace) at 4°C overnight. The cleared supernatant containing the solubilized MBP-ELIC fusion protein was purified by affinity chromatography on amylose resin (New England Biolabs). Column-bound ELIC was cleaved off by 3CV protease in the presence of 1 mM EDTA + 1 mM DTT at 4°C overnight. A final purification step was carried out on a Superdex 200 Increase 10/300 GL column (GE Healthcare) equilibrated with buffer containing 10 mM Na-phosphate pH 8.0, 150 mM NaCl, and 0.15% n-undecyl-β-D-maltoside (UDM, Anatrace). Peak fractions containing pentameric ELIC were pooled, concentrated to ~10 mg/mL and relipidated with 0.5 mg/mL *E. coli* lipids (Avanti Polar Lipids). Nanobodies were added at a 20% molar excess calculated for monomers and incubated at room temperature 2 hr prior to setting up crystallization screens with a Mosquito liquid handling robot (TTP Labtech). Crystals for the ELIC complex with PAM-Nb grew at

room temperature in the presence of 0.1 M GABA, 0.2 M Ca(OAc)$_2$, 0.1 M MES buffer pH 6.5% and 10% PEG8000. Crystals for the ELIC complex with NAM-Nb grew at room temperature in the presence of 0.1 M Na$_2$SO$_4$, 0.1 M bis-trispropane pH 8.5% and 10% PEG3350. Crystals were harvested after adding cryo-protectant containing mother liquor gradually supplemented with up to 25% glycerol in 5% increments. Crystals were then plunged into liquid nitrogen and stored in a dewar for transport to the synchrotron.

## Structure determination of ELIC-nanobody complexes

Diffraction data for the ELIC+PAM-Nb structure were collected at the PROXIMA 1 beamline of the SOLEIL synchrotron (Gif-sur-Yvette, France). Diffraction data for the ELIC+NAM-Nb structure were collected at the X06A beamline of the Swiss Light Source (Villigen, Switzerland). Both structures were solved by molecular replacement with Phaser in the CCP4 suite (*Winn et al., 2011*) using the published structures for the ELIC pentamer (pdb 2vl0) and a GPCR nanobody (pdb 3p0g) as search templates. For the ELIC+PAM-Nb structure, the asymmetric unit contains one ELIC pentamer with five nanobodies bound (one to each subunit). For the ELIC+NAM-Nb structure, the asymmetric unit contains two ELIC pentamers with one nanobody bound to each pentamer. The electron density for one of these nanobody molecules is not well defined, suggesting partial occupancy at this pentamer, and therefore the atom occupancies for this nanobody were manually set to 40% during structure refinement. The data set for the ELIC+NAM-Nb was anisotropic with data extending to ~3.15 Å in the best direction and ~3.5 Å in the worst. To correct for anisotropy the unmerged reflections from XDS (*Kabsch, 2010*) were uploaded to the STARANISO server (*Tickle et al., 2018*) and automatically processed using CC1/2 > 30 and I/σ >2 as resolution cut-off criteria. The merged and scaled data set from this procedure extends to a resolution of 3.25 Å and the statistics produced by the STARANSIO server are shown in the crystallographic table (*Supplementary file 1*). Structures were improved by iterative rounds of manual rebuilding in Coot (*Emsley et al., 2010*) and automated refinement in Buster (*Smart et al., 2012*) or Refmac (*Winn et al., 2011*). Structure validation was carried out in PDB-REDO (*Joosten et al., 2014*) and MolProbity (*Chen et al., 2010*). Figures were prepared with PyMOL (Schrödinger) and UCSF-Chimera (*Pettersen et al., 2004*). Pore radius profiles were made using HOLE (*Smart et al., 1996*) and CAVER (*Jurcik et al., 2018*). Simulated annealing omit maps were calculated in PHENIX (*Adams et al., 2010*) and are shown for the nanobody-ELIC interaction region (*Figure 1—figure supplements 2–3*).

## Cysteine-scanning mutagenesis and voltage-clamp recordings of 5-HT$_{3A}$R

Stage V-VI *Xenopus* oocytes were purchased from Ecocyte (Germany) and stored in ND-96 (96 mM NaCl, 2 mM KCl, 1.8 mM CaCl$_2$, 1 mM MgCl$_2$, 5 mM HEPES, pH 7.5) containing 2.5 mM sodium pyruvate, 50 mM gentamicin and 0.7 mM theophylline. cDNA encoding human 5-HT$_{3A}$R was cloned into the pGEM-HE expression plasmid (*Liman et al., 1992*). Mutants were engineered using the QuikChange mutagenesis kit (Agilent) and confirmed by sequencing. cRNA was in vitro transcribed from linearized pGEM-HE cDNA template using the mMessage mMachine T7 Transcription kit (ThermoFisher). Oocytes were injected with 50 nl of ~400 ng/μl cRNA, and currents were recorded 18–48 hr post-injection. 5-HT$_{3A}$R current recordings were obtained using a Roboocyte voltage-clamp system (Multi Channel systems) at a constant voltage clamp of −60 mV. Oocytes were perfused with ND-96 with no added calcium, and 5-HT (creatinine sulphate complex, Sigma) was diluted in this media. Oocytes were tested at 10 μM 5-HT before obtaining concentration-response curves. MTSEA-biotin (Biotium) was diluted immediately prior to application into calcium-free ND-96 solution at a concentration of 2 mM from a stock solution of 500 mM in DMSO. Analysis and curve fitting was performed using Prism v4.03 (GraphPad Software). Concentration-response data for each oocyte were normalized to the maximum current for that oocyte. Data are presented as the mean ± standard error of the mean (SEM) with the raw data points overlaid as a dot plot in the relevant figures.

## Transparent reporting statements

All *Xenopus* electrophysiology experiments were repeated three to eight times. The number of 'n' is mentioned in the relevant sections of the main text. We define each separate oocyte recording as a biological repeat. No data were excluded, unless the oocyte gave no detectable current. All

electrophysiology experiments were conducted on automated devices, either the HiClamp or the Roboocyte, so essentially there was no human bias in recording of these data. Data are presented as the mean ± standard error of the mean (SEM) with the raw data points overlaid as a dot plot. Statistical comparison between groups of data was performed using an unpaired two-tailed $t$ test or an ANOVA followed by a Dunnetts multiple comparisons test as appropriate; the significance value p is mentioned in the relevant sections of the manuscript.

The X-ray diffraction data sets were collected from single crystals and typically the data set with the highest resolution was used for structural elucidation. Equivalent reflection data were recorded multiple times in agreement with the rotational symmetry of the crystal packing. The relevant data multiplicity value for each data set is mentioned in the crystallographic table (*Supplementary file 1*). All aspects of X-ray data collection, integration, scaling and merging were fully automated so human bias was excluded. No data were excluded.

## Acknowledgements

We thank beamline staff at the SOLEIL synchrotron and Swiss Light Source for assistance with data collection. SBO/IWT-project 1200261 and FWO-project G.0762.13 were awarded to JS and CU. Additional support was from KU Leuven OT/13/095, C32/16/035 and C14/17/093 to CU.

## Additional information

### Funding

| Funder | Grant reference number | Author |
|---|---|---|
| Agentschap Innoveren en Ondernemen | 1200261 | Jan Steyaert<br>Chris Ulens |
| Fonds Wetenschappelijk Onderzoek | G.0762.13 | Jan Steyaert<br>Chris Ulens |
| KU Leuven | OT/13/095 | Chris Ulens |
| KU Leuven | C32/16/035 | Chris Ulens |
| KU Leuven | C14/17/093 | Chris Ulens |
| Instruct-ERIC | | Jan Steyaert |
| National Institutes of Health | DA047325 | Ryan E Hibbs |
| National Institutes of Health | DA042072 | Ryan E Hibbs |
| National Institutes of Health | NS095899 | Ryan E Hibbs |

The funders had no role in study design, data collection and interpretation, or the decision to submit the work for publication.

### Author contributions

Marijke Brams, Formal analysis, Methodology, Writing - original draft, Writing - review and editing; Cedric Govaerts, Data curation, Formal analysis, Validation, Writing - original draft, Writing - review and editing; Kumiko Kambara, Genevieve L Evans, Formal analysis, Validation, Visualization, Writing - original draft, Writing - review and editing; Kerry L Price, Data curation, Formal analysis, Visualization, Methodology, Writing - original draft, Writing - review and editing; Radovan Spurny, Data curation, Formal analysis, Validation, Visualization, Methodology; Anant Gharpure, Conceptualization, Formal analysis, Validation, Visualization, Methodology, Writing - original draft, Writing - review and editing; Els Pardon, Resources, Formal analysis, Validation, Methodology, Writing - original draft, Writing - review and editing; Daniel Bertrand, Data curation, Software, Formal analysis, Validation, Visualization, Methodology, Writing - original draft, Writing - review and editing; Sarah CR Lummis, Conceptualization, Data curation, Formal analysis, Funding acquisition, Validation, Visualization, Methodology, Writing - original draft, Project administration, Writing - review and editing; Ryan E Hibbs, Conceptualization, Data curation, Formal analysis, Validation, Visualization, Methodology, Writing - original draft, Writing - review and editing; Jan Steyaert, Conceptualization, Data curation,

Formal analysis, Funding acquisition, Methodology, Writing - original draft, Writing - review and editing; Chris Ulens, Conceptualization, Resources, Data curation, Formal analysis, Supervision, Funding acquisition, Validation, Investigation, Visualization, Methodology, Writing - original draft, Project administration, Writing - review and editing

### Author ORCIDs
Anant Gharpure  http://orcid.org/0000-0002-4458-359X
Genevieve L Evans  http://orcid.org/0000-0002-8612-9539
Jan Steyaert  http://orcid.org/0000-0002-3825-874X
Chris Ulens  https://orcid.org/0000-0002-8202-5281

### Decision letter and Author response
Decision letter https://doi.org/10.7554/eLife.51511.sa1
Author response https://doi.org/10.7554/eLife.51511.sa2

## Additional files

### Supplementary files
- Supplementary file 1. Crystallographic and refinement statistics.
- Supplementary file 2. Structures used for structure-based superposition and $\Omega$-loop analysis.
- Supplementary file 3. Effects of MTSEA-biotin (MB) application on WT and mutant 5-HT$_3$R.
- Transparent reporting form

### Data availability
Atomic coordinates and structure factors have been deposited with the Protein Data Bank under accession numbers 6SSI for the ELIC+PAM-Nb structure and 6SSP for the ELIC+NAM-Nb structure. The raw X-ray diffraction images for both data sets have been deposited on Dryad under DOI https://doi.org/10.5061/dryad.pv4097s.

The following datasets were generated:

| Author(s) | Year | Dataset title | Dataset URL | Database and Identifier |
|---|---|---|---|---|
| Chris Ulens, Marijke Brams, Genevieve L Evans, Radovan Spurny, Cedric Govaerts, Els Pardon, Jan Steyaert | 2020 | ELIC+PAM-Nb | https://www.rcsb.org/structure/6SSI | RCSB Protein Data Bank, 6SSI |
| Chris Ulens, Marijke Brams, Genevieve L Evans, Radovan Spurny, Cedric Govaerts, Els Pardon, Jan Steyaert | 2020 | ELIC+NAM-Nb | https://www.rcsb.org/structure/6SSP | RCSB Protein Data Bank, 6SSP |
| Chris Ulens, Marijke Brams, Genevieve L Evans, Radovan Spurny, Cedric Govaerts, Els Pardon, Jan Steyaert | 2020 | Data from: Modulation of the Erwinia ligand-gated ion channel (ELIC) and the 5-HT3 receptor via a common vestibule site | http://dx.doi.org/10.5061/dryad.pv4097s | Dryad Digital Repository, 10.5061/dryad.pv4097s |

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
