## [Decision Letter]

**Acceptance summary:**

Using a combination of X-ray crystallography and targeted mutagenesis combined with functional assays, this manuscript identifies two nanobodies that allosterically modulate the bacterial pentameric ligand-gated ion channel (pLGIC), ELIC, and examines the structural mechanisms underlying their functional effects. One nanobody acts as a positive allosteric modulator (PAM) and the other a negative allosteric modulator (NAM). X-ray crystal structures reveal that the NAM-nanobody binds to a novel site in the upper extracellular domain vestibule of ELIC. The PAM-nanobody binding site overlaps with a previously-identified allosteric drug site in the related nicotinic acetylcholine receptor. Guided by the structures, the authors examine the vestibule binding site in the related serotonin type 3 receptor and show that perturbing this region (via covalent modification and/or mutation) alters channel activation suggesting the site might represent a new region to target for drug development. The data provide new information about mechanisms underlying allosteric modulation of pLGICs and will be of interest to many in the field.

**Decision letter after peer review:**

Thank you for submitting your article "A functionally conserved mechanism of modulation via a vestibule site in pentameric ligand-gated ion channels" for consideration by *eLife*. Your article has been reviewed by three peer reviewers, and the evaluation has been overseen by a Reviewing Editor and Kenton Swartz as the Senior Editor. The following individuals involved in review of your submission have agreed to reveal their identity: Alexander Sobolevsky (Reviewer #1); Alexandru Radu Aricescu (Reviewer #3).

The reviewers have discussed the reviews with one another and the Reviewing Editor has drafted this decision to help you prepare a revised submission.

The major results and conclusions are:

1) Identified two nanobodies that bind to ELIC: one that acts as a positive allosteric modulator (PAM) and the other as a negative allosteric modulator (NAM).

2) Solved high-resolution structures of ELIC bound to PAM-nanobody and to NAM-nanobody.

3) The NAM-nanobody binds to a novel site in the upper ECD vestibule of ELIC.

4) Covalently linking a bulky reagent MTSE-biotin into a pocket in the vestibule region in the 5HT3R modulates serotonin-evoked currents suggesting the upper ECD vestibule might represent a new target for developing drugs.

The authors' conclusions are supported by the experimental data. The manuscript is well written and offers new insights into a novel intra-vestibule binding site on pentameric ligand-gated ion channels (pLGICs).

Addressing the following essential revisions will strengthen the manuscript.

Essential Revisions:

1) Revise title so it is less general and does not overstate manuscript's main conclusion. Abstract should be revised and describe specific experiments done and results of these experiments. Do not generalize results to all pLGICs.

2) Previous work from Hassaine et al., 2014 (5HT3R) and Miller et al., 2018 (bioRxiv) have reported that nanobodies modulate pLGIC function – please mention this point and add references.

3) A modulatory drug vestibule site may not be a general feature of all pLGICs. For GABA-A receptors, vestibule is occupied by glycans and thus is not accessible. Authors need to be careful not to generalize the conclusions drawn from their work to all pLGIC members.

4) Authors need a more thorough analysis/discussion of residues involved in positive versus negative modulation. Figure 2 needs to be revised and/or an additional supplementary figure should be included to highlight the residue side chains that interact with the nanobodies versus small ligands. Compare and contrast nanobody interaction residues with CU2017 and flurazepam contact residues. Providing dose-response curves in the presence of the nanobodies would provide additional mechanistic insight into their mode of action.

5) Please include/discuss whether there any examples of allosteric modulators that bind in the vestibule of other members of the pLGIC superfamily e.g. nAChRs, GABARs, GlyRs, 5HT3Rs. Could pLGICs with omega-in and omega-out conformations also be modulated via the vestibule?

6) Include data showing effects of MTSEA-biotin on wild-type ELIC (control). Provide rationale for using MTSEA-biotin as the modifying reagent. Summarize SCAM results for the tested cysteine mutant receptors in a table.

7) NAM nanobody blocks vestibule but only 35% of the GABA evoked current can be blocked. Additional discussion is needed to support idea that current enters via fenestrations at subunit interfaces. Show on ELIC structure where this could take place along with assessments of ion sizes in relation to fenestration/tunnel dimensions. Alternative mechanisms are possible and should be discussed. Could NAM-NB binding restrict flexibility in the ECD, thereby reducing the efficacy of the orthosteric agonist? Is it possible that the NAM-Nb-bound vestibule is occluded in the apo/desensitized states but not in an open state, due to an agonist-induced conformational change?

8) The authors' need to elaborate more about omega loop and how they envision this region of the protein regulating channel activation and drug modulation. Are conformations of the omega loop different in GluCl apo, open, desensitized conformations, GLIC apo, open, desensitized structures, and in different GABA-A receptor conformations. Do they imagine this loop intrinsically flexible? Do you think only omega-open channels are drug targets? Previous studies on nACHR (Chakrapani, Bailey and Auerbach, 2003) and GABA-A receptors (Hanek, Lester and Dougherty, 2010; Ventatachalan and Czajkowski, 2012) have mutated this region and should be discussed.

9) Flurazepam is a 'classic' benzodiazepine targeting an extracellular binding site between α- and γ-subunits of GABA-A receptors, quite far from the (occluded) Ω-out vestibular site. It will be useful for the general reader, if you mention that flurazepam has more than one binding site in the pLGIC family. It should be clearly stated that flurazepam site in ELIC is not the site that corresponds to its action on the eukaryotic GABAR.

10) L151C modification by MTSEA-biotin has large functional effects. Can the authors comment on why? The cysteine substitution of L151 alone has large effects on function. Please discuss.

11) Figures 1 and 4 – please include the number of experiments performed for each of the dose response curves. Data are mean +/- SEM?

12) NAM nanobody site in ELIC does not overlap with previously identified flurazepam site. NAM nanobody site sits above the cavity that binds flurazepam. Please comment on and discuss.

13) Previous studies have shown that targeted modification of introduced cysteine residues by sulfhydryl reactive reagents throughout the ECD and TMD of many different pLGICs alter channel activation and/or drug modulation. It is unlikely that all of these sites are targets for drug development. The SCAM experiments are interesting and demonstrate that structural perturbations at these residues effect receptor activation but authors need to be careful about interpreting the data as evidence for a conserved allosteric binding site in the vestibule.

---

## [Author Response]

Essential Revisions:1) Revise title so it is less general and does not overstate manuscript's main conclusion. Abstract should be revised and describe specific experiments done and results of these experiments. Do not generalize results to all pLGICs.

We have revised the title and Abstract as requested.

2) Previous work from Hassaine et al., 2014 (5HT3R) and Miller et al., 2018 (bioRxiv) have reported that nanobodies modulate pLGIC function – please mention this point and add references.

These references are now cited and discussed in the subsection “Conclusion”.

3) A modulatory drug vestibule site may not be a general feature of all pLGICs. For GABA-A receptors, vestibule is occupied by glycans and thus is not accessible. Authors need to be careful not to generalize the conclusions drawn from their work to all pLGIC members.

We agree that modulation via the vestibule site may not be a general feature of pLGICs and have been more cautious in this revised version, where we have also mentioned N-linked glycan access restriction of the vestibule in GABA_A_ receptors (subsection “Subtype-dependent vestibule site access in different prokaryote and eukaryote receptors”, last paragraph).

4) Authors need a more thorough analysis/discussion of residues involved in positive versus negative modulation. Figure 2 needs to be revised and/or an additional supplementary figure should be included to highlight the residue side chains that interact with the nanobodies versus small ligands. Compare and contrast nanobody interaction residues with CU2017 and flurazepam contact residues. Providing dose-response curves in the presence of the nanobodies would provide additional mechanistic insight into their mode of action.

These are good suggestions. We have now analyzed residues involved in PAM-Nb and NAM-Nb interactions in more detail and compared them to CU2017 (NAM) and flurazepam (PAM) interactions, respectively. This is discussed in the third paragraph of the subsection “Crystal structures of ELIC in complex with a PAM- or NAM-type nanobody”, and new Figure 2—figure supplement 1-2.

We have also determined GABA concentration-activation curves in the presence of nanobodies as suggested, and indeed this has revealed further mechanistic insight consistent with their mode of action as allosteric modulators. This is discussed in the last paragraph of the subsection “Identification of nanobodies active as allosteric modulators on ELIC”, and new Figure 1—figure supplement 1.

5) Please include/discuss whether there any examples of allosteric modulators that bind in the vestibule of other members of the pLGIC superfamily e.g. nAChRs, GABARs, GlyRs, 5HT3Rs. Could pLGICs with omega-in and omega-out conformations also be modulated via the vestibule?

We now describe several known examples of vestibule site modulators in different members of the pLGIC superfamily in the last paragraph of the subsection “Cysteine-scanning mutagenesis in the vestibule site of the human 5-HT3A receptor”.

Regarding the modulation of Ω-in and Ω-out conformations, our structural analysis shows that vestibule access for these is restricted. It is perhaps possible (as we discuss in the revised manuscript) that ligand-induced conformational changes of the Ω-loop could occur, permitting vestibule site access, but in the absence of any evidence this is too speculative to consider in any detail.

6) Include data showing effects of MTSEA-biotin on wild-type ELIC (control). Provide rationale for using MTSEA-biotin as the modifying reagent. Summarize SCAM results for the tested cysteine mutant receptors in a table.

We assume this remark relates to the 5-HT_3_ receptor, not to ELIC. The data are now summarized in a new Supplementary file 3 along with data from wild type 5-HT_3_R (where the lack of effect of MTSEA-biotin was previously demonstrated in Thompson et al., 2011). We also provide a rationale for using MTSEA-biotin in the first paragraph of the subsection “Cysteine-scanning mutagenesis in the vestibule site of the human 5-HT3A receptor”.

7) NAM nanobody blocks vestibule but only 35% of the GABA evoked current can be blocked. Additional discussion is needed to support idea that current enters via fenestrations at subunit interfaces. Show on ELIC structure where this could take place along with assessments of ion sizes in relation to fenestration/tunnel dimensions. Alternative mechanisms are possible and should be discussed. Could NAM-NB binding restrict flexibility in the ECD, thereby reducing the efficacy of the orthosteric agonist? Is it possible that the NAM-Nb-bound vestibule is occluded in the apo/desensitized states but not in an open state, due to an agonist-induced conformational change?

We analyzed ELIC structures in apo as well as nanobody-bound structures and found lateral fenestrations at the same location as in the β3 GABA_A_ receptor structure (PDB 4COF), although slightly smaller in radius. This provides additional support for an alternate ion pathway in the NAM-Nb bound ELIC state. This analysis was conducted using the program CAVER, which also revealed a partially-blocked ion conduction pathway along the extracellular vestibule entrance. This result was also confirmed using the program HOLE and could provide an additional explanation for the partial reduction of the GABA-evoked current in the presence of NAM-Nb. Finally, we also agree with the reviewer that the NAM-Nb could restrict flexibility in the ECD, thereby limiting a gating transition to an open/active conformation. This is now discussed in the last paragraph of the subsection “Crystal structures of ELIC in complex with a PAM- or NAM-type nanobody”.

We have updated the pore radius profiles throughout the study with the results from HOLE and CAVER, as the program CHAP appears to produce deviating results for the vestibule entrance (see Author response image 1).

8) The authors' need to elaborate more about omega loop and how they envision this region of the protein regulating channel activation and drug modulation. Are conformations of the omega loop different in GluCl apo, open, desensitized conformations, GLIC apo, open, desensitized structures, and in different GABA-A receptor conformations. Do they imagine this loop intrinsically flexible? Do you think only omega-open channels are drug targets? Previous studies on nACHR (Chakrapani, Bailey and Auerbach, 2003) and GABA-A receptors (Hanek, Lester and Dougherty, 2010; Ventatachalan and Czajkowski, 2012) have mutated this region and should be discussed.

We have elaborated on this in the revised manuscript. A comprehensive analysis of all pLGIC structures available to date, including >100 GLIC structures, revealed no clear correlation between the Ω-loop conformation and apo, open, desensitized structures in GluCl or GLIC, or any other pLGICs. Analysis of the average B-factor per residue, which is used as an indicator of vibrational movement, did reveal that in certain structures the Ω-loop has enhanced B-factors relative to other parts of the structure. Some of the most striking examples are the Ω-in loop of the a3 subunit in the a3b4 nAChR (PDB code 6PV7, desensitized state) (Gharpure et al., 2019), the Ω-out loop in POPC-bound GluCl (PDB code 4TNW, lipid-modulated state) (Althoff et al., 2014) and the Ω-open loop in GLIC T25’A (PDB code 4LMJ, liganded closed state) (Gonzalez-Gutierrez et al., 2013). These examples also represent possible intermediate or end states of the gating cycle, suggesting that the Ω-loop could show enhanced movement during the gating reaction. This is consistent with the very relevant studies mentioned by the reviewers, which are now cited and discussed (subsection “Subtype-dependent vestibule site access in different prokaryote and eukaryote receptors”).

9) Flurazepam is a 'classic' benzodiazepine targeting an extracellular binding site between α- and γ-subunits of GABA-A receptors, quite far from the (occluded) Ω-out vestibular site. It will be useful for the general reader, if you mention that flurazepam has more than one binding site in the pLGIC family. It should be clearly stated that flurazepam site in ELIC is not the site that corresponds to its action on the eukaryotic GABAR.

We now state explicitly that flurazepam has multiple binding sites and that the vestibule site in ELIC does not correspond to the high affinity binding benzodiazepine binding site in human GABA_A_Rs. See the third paragraph of the subsection “Crystal structures of ELIC in complex with a PAM- or NAM-type nanobody”.

10) L151C modification by MTSEA-biotin has large functional effects. Can the authors comment on why? The cysteine substitution of L151 alone has large effects on function. Please discuss.

This is indeed interesting as the L151C mutation alone significantly increases the EC_50_-value compared to wild type (EC_50_ 98 µM versus 1.8 µM) suggesting that this mutation impacts function. In the revised manuscript we suggest it might be because L151 contributes to a hydrophobic patch in the center of the subunit (it is < 4 Å from I100 and L131) and this – or the mutation itself – could affect the adjacent Cys-loop and loop B. This is discussed in the last paragraph of the subsection “Cysteine-scanning mutagenesis in the vestibule site of the human 5-HT3A receptor”.

11) Figures 1 and 4 – please include the number of experiments performed for each of the dose response curves. Data are mean +/- SEM?

The number of experiments is now mentioned in the text as well as the summarizing SCAM table (new Supplementary file 3). Data are presented as the mean ± standard error of the mean (SEM) with the raw data points overlaid as a dot plot in the relevant figures. We now mention this specifically in the last paragraphs of the subsections “Identification of nanobodies active as allosteric modulators on ELIC” (Figure 1), “Cysteine-scanning mutagenesis in the vestibule site of the human 5-HT3A receptor” (Figure 4), the Materials and methods section, and in the “Transparent Reporting Form”.

12) NAM nanobody site in ELIC does not overlap with previously identified flurazepam site. NAM nanobody site sits above the cavity that binds flurazepam. Please comment on and discuss.

The NAM-Nb interaction site is the a’1-helix in ELIC (N60-N69), which forms the top of the vestibule binding site. The flurazepam site and NAM-Nb site are adjacent to each other, see new Figure 2—figure supplement 2. This is discussed in the third paragraph of the subsection “Crystal structures of ELIC in complex with a PAM- or NAM-type nanobody”.

13) Previous studies have shown that targeted modification of introduced cysteine residues by sulfhydryl reactive reagents throughout the ECD and TMD of many different pLGICs alter channel activation and/or drug modulation. It is unlikely that all of these sites are targets for drug development. The SCAM experiments are interesting and demonstrate that structural perturbations at these residues effect receptor activation but authors need to be careful about interpreting the data as evidence for a conserved allosteric binding site in the vestibule.

The reviewer is correct that SCAM experiments should be interpreted with caution, but at the structural level the 5-HT_3_ receptor has an accessible vestibule site that very closely resembles ELIC. At the functional level, the mutagenesis and SCAM experiments also closely mimic the potentiation of ELIC by flurazepam. Combined, these data suggest that receptors with an accessible vestibule such as the 5-HT_3_ receptor could be targets for drug development.